



# Mapping snow depth and volume at the alpine watershed scale from aerial imagery using Structure from Motion

Joachim Meyer[1], S. McKenzie Skiles[1], Jeffrey Deems[2], Kat Bormann[3], David Shean[4]

[1] Department of Geography, University of Utah, Salt Lake City, UT, USA
[2] National Snow and Ice Data Center, Boulder, CO, USA
[3] Airborne Snow Observatories, Inc., Mammoth Lakes, CA, USA
[4] Dept. of Civil and Environmental Engineering, University of Washington, Seattle, WA, USA

*Correspondence to*: Joachim Meyer (j.meyer@utah.edu)

## Abstract

Time series mapping of water held as snow in the mountains at global scales is an unsolved challenge to date. In a few locations, lidar-based airborne campaigns have been used to provide valuable data sets that capture snow distribution in near real-time over multiple seasons. Here, an alternative method is presented to map snow depth and quantify snow volume using aerial images and Structure from Motion (SfM) photogrammetry over an alpine watershed (300 km$^2$). The results were compared to the lidar-derived snow depth measurements from the Airborne Snow Observatory, collected simultaneously.

Where snow was mapped by both ASO and SfM, the depths compared well, with a mean difference of 0.01 m, NMAD of 0.22 m, and snow volume agreement (difference 1.26%). ASO though, mapped a larger snow area relative to SfM, with SfM missing ~14% of total snow volume as a result. Analyzing the SfM reconstruction errors shows that challenges for photogrammetry remain in vegetated areas, over shallow snow (< 1 m), and slope angles over 50 degrees. Our results indicate that capturing large scale snow depth and volume with airborne images and photogrammetry could be an additional

viable resource for understanding and monitoring snow water resources in certain environments.



## 1. Introduction

Snow depth and snow water equivalent are essential observables for many water resource applications. In alpine environments, snow depth is traditionally measured continuously at instrumented sites or periodically along transects due to the complexity of the terrain. These long-term records are valuable but tend to be located at mid-elevations index sites that are accessible and hold snow for longer than the surrounding terrain. This limited spatial coverage leaves a poor understanding of snow depth distributions in the mountains, particularly at high elevations. The gap can be addressed by mapping snow depth differentially using remotely sensed surface elevation products (Deems et al 2013). Snow depth can be estimated with a pixel-wise calculation on raster-based products that subtracts snow-free elevations from snow-on elevations over a target area.

This principle has been demonstrated from several remote sensing platforms, spanning a range of spatial resolution and coverage, repeat intervals, and snow depth accuracy. The Ice, Cloud, and Land Elevation Satellite (ICESat) was a laser altimeter that could map snow depths along swaths with sub-meter accuracy (Treichler & Kääb, 2017), but the data are of limited utility for mountain regions due to the low temporal resolution and large ground footprint (70 m). Satellite stereo photogrammetry derived DEMs, such as those from WorldView or Pléiades, have the potential for higher spatial (< 1 m) and temporal resolution (Shean et al., 2016, McGrath et al., 2019, Deschamps-Berger et al., 2020). Limitations, though, include reduced accuracy in complex terrain, 10–50 cm over shallow slopes (<10°; Shean et al., 2016), and data gaps when the target area is obstructed, by clouds for example (Shaw et al., 2020). With no current space-borne platform providing the combination of high temporal/spatial resolution and high accuracy required for distributed snow mapping, airborne campaigns have been established to address these needs. For example, the Airborne Snow Observatory (ASO), combining a lidar and imaging spectrometer platform, delivers time-series of snow depth maps at 3 m resolution with centimeter accuracy in select watersheds primarily in the California Sierra Nevada and Colorado Rocky Mountains (Painter et al., 2016). Although smaller in spatial extent and periodic, it has been shown that Structure from Motion (SfM) photogrammetry using imagery from Remotely Piloted Aircraft System (RPAS) can map snow depth at sub-decimeter resolution, while maintaining centimeter accuracy for areas up to alpine catchments size (Bühler et al., 2016; Harder et al., 2016; Schirmer & Pomeroy, 2020).

Airborne platforms have important limitations, such as expense and logistics for a piloted aircraft, weather restrictions for remotely piloted and piloted aircraft, and limited areal coverage with RPASs due to battery life. Limitations lead to significantly smaller footprints and less consistent coverage relative to space-borne platforms. Still, the ability to acquire high resolution/high accuracy data sets on-demand, over any desired target area, makes airborne campaigns an essential tool for both water management operations and research in alpine environments. The resulting data sets have expanded our knowledge of snow science in watersheds (Behrangi et al., 2018; Brandt et al., 2020; Hedrick et al., 2018, Zheng et al., 2019) and are now well established as relevant data sources. In particular, the RPAS-SfM studies have seen a recent gain in





popularity due to high affordability using consumer-grade cameras that can deliver highly accurate data sets (Gaffey & Bhardwaj, 2020).

This paper evaluates the ability of SfM to map snow depth distributions over a watershed with high-resolution imagery captured by a piloted aircraft relative to coincidentally collected lidar-based retrievals. There have been exceptions (Nolan et al., 2015; Eberhard et al., 2020), but to date, snow depth mapping from high altitude piloted aircraft has been lidar-based, while low altitude RPAS platforms have been primarily SfM based. *Meyer & Skiles* (2019) showed that accurate DEMs can be generated from imagery collected from piloted aircraft over bright snow surfaces using SfM. Building upon this work, we

show that SfM DEMs can be used to differentially calculate snow depths and corresponding snow volume over a relatively large alpine watershed (300 km$^2$) at scales commensurate with airborne lidar-based applications. This comparison demonstrates that SfM is a reliable remote sensing technique for large-scale DEM reconstruction and differential volume mapping in complex terrain. Additionally, the coincidental collection with lidar provides a unique opportunity to further expand our understanding of the strengths and weaknesses of applying photogrammetric-based techniques with areal snow

observations.

## 2. Study area

The East River watershed, located northeast of Crested Butte, CO, lies within the broader Upper Gunnison watershed. It encompasses the long running Rocky

Mountain Biological Laboratory and a portion of Crested Butte Mountain Resort. The East River is one of two primary tributaries of the Gunnison River, which itself discharges into the Colorado River. The watershed is an estimated 300 km$^2$ in size and has an average elevation of

3266 m and vertical relief of 1420 m (Hubbard et al., 2018). The vegetation varies across the elevation ranges and includes brush and grass land, aspen and mixed conifer, and alpine meadows. The East River was designated as a Scientific Focus Area in 2016, supported by the US-DOE

Biological and Environmental Research Subsurface Biogeochemistry Program. The Airborne Snow Observatory flights, and subsequent data processing, were funded by the state of Colorado to map snow distribution patterns and support water supply forecast improvements.

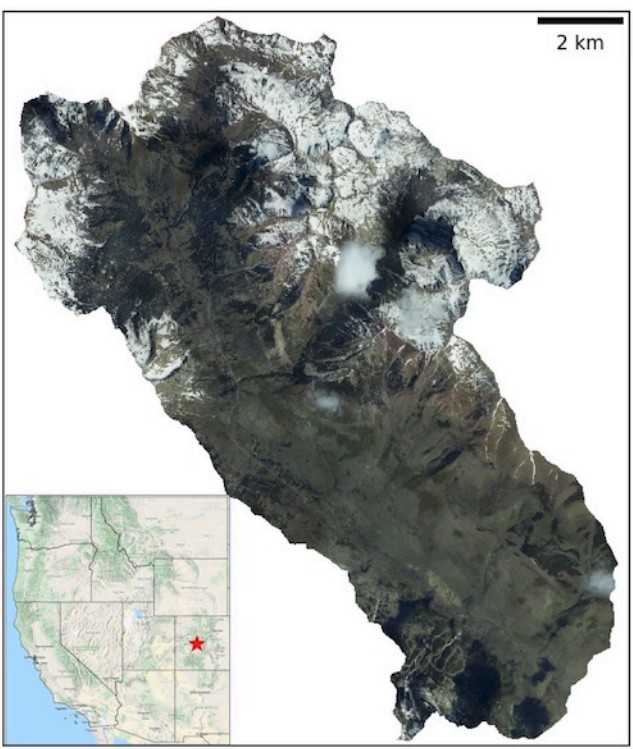

**Figure 1 - Areal overview of the East River watershed and its location shown relative to the Western United States. (Insert map base layer: Map data ©2021 Google)**



## 3. Data

The ASO flights took place on the 24 May 2018 for the snow-on scene and 12 September 2019 for the snow-free. Flight patterns were almost identical on both days, covering the area with a 50% overlap lawn-mower pattern. Flight altitude varied slightly between the two flights, where the May flight was 6400 m above sea level, and the September flight was at 6100 m. There was also a difference in flight line orientation between the two dates, with the May flights in a North-South direction and the September flights in a Northwest -Southeast direction. The orientation for the flight lines during the snow season were selected based on lighting conditions for the ASO imaging spectrometer and flight efficiency, and no direct considerations for the camera are given.

The camera used by ASO is mounted inside the lidar instrument, which creates identical view perspectives between the lidar scanner and the camera to the ground surface. Each image has dimensions of $10,328 \times 7,760$ pixels with a 16-bit color depth and size of 5.2 micron for an individual pixel. Underlying hardware consisted of a medium format Phase One iXU 180-R CCD sensor camera with a Rodenstock 50 mm HR Digaron-W wide-angle view lens. The recording interval for the camera was twelve seconds for the snow-free flight, which resulted in 287 images, and six seconds for the snow-on flight resulting in 582 images. The average ground sample distance (GSD) was 0.31 m/pixel for the snow-on and 0.28 m/pixel for the snow-free images. An overview for both collections is shown in Table 1.

For quality assessment of the measured depth by SfM, we used the publicly available snow depth product by ASO, which is published through the National Snow and Ice Data Center. ASO uses the identical difference principle to calculate depth, where snow-on values are subtracted with the snow-free. More technical details on the ASO platform and the final output product's processing steps can be found in Painter et al., 2016.

**Table 1 - Flight parameters for snow-on and snow-free recording.**

|  | *24 May (snow-on)* | *12 September (snow-free)* |
|---|---|---|
| Flight pattern | Single overlap, lawn-mower | Single overlap, lawn-mower |
| Flight line orientation | North-South | Northwest-Southeast |
| Flight altitude (above sea level) | 6400 m | 6100 m |
| Camera recording interval | 12s | 6s |
| Number of images | 287 | 582 |
| Mean GSD | 0.31 m/pixel | 0.28 m/pixel |

## 4. Methods

### 4.1 Image Processing

The camera images from the ASO survey were processed using Agisoft Metaphase (version 1.6.2) along with associated geo-location and orientation data from the airplane global navigation satellite system and inertial measurement unit.



Metashape was used for feature matching, image alignment, and dense point cloud creation. We refer the interested readers
for more technical details on data preparation and settings for Metashape to the workflow in *Meyer & Skiles* (2019).

### 4.2 Co-Registration

After the SfM snow-free and snow-on point clouds were generated, a reference lidar elevation data set ensured the closest
alignment of surface models through co-registration. This minimizes relative geo-location error, providing improved
accuracy for DEM difference products. The co-registration was performed using the Ames Stereo Pipeline (ASP; version
2.6.3), which internally uses the iterative closest point algorithm to determine the difference between two point clouds
(Shean et al., 2016, Beyer et al., 2018). The reference point cloud consisted of control surfaces from the ASO snow-on
acquisition flight. Control surfaces were identified from the ASO imaging spectrometer classification and are consistent
elevation across time, such as exposed bedrock or roads. The control surfaces were additionally refined by removing any
areas that had snow in the ASO snow depth product and any slopes steeper than 50 degrees (Shaw et al., 2020). The
bounding box for the reference DEM was extended beyond the watershed boundaries to increase the available area for co-
registration. An added advantage of co-registering of the SfM point clouds to the ASO lidar point cloud was the implicit
alignment with the ASO snow depth product.

The co-registered point clouds were converted to a gridded raster product (GeoTIFF) with 1 m resolution using the Point
Data Abstraction Library (PDAL, Contributors, 2018), which provides the inverse distance weighting (IDW) algorithm for
interpolation. The IDW algorithm can be applied with a point density of multiple points per square meter (Guo et al., 2010),
and both SfM clouds had sufficient density for its application at the 1 m resolution. In addition to the resolution and
algorithm, PDAL was also used to clip the outputs to identical bounding boxes and transform to matching projection (WGS
84 / UTM zone 13N; EPSG 32613). The final step was calculating the SfM snow depth by taking the pixel wise difference in
surface elevation between the snow-on and snow-free DEMs.

### 4.3 Comparison


The snow depth (SD) values from SfM were compared to the ASO snow depth map by treating ASO as the reference, since
snow depth mapping with lidar is the more established method. ASO distributes its snow depth products at 3 m resolution
and as compromise between the possible higher SfM and available ASO resolutions, we compared the products at the 1 m
resolution by down sampling the 3 m ASO snow depths.
The SfM snow depths were compared to ASO snow depths using the full domain mean, median, and standard deviation for
each data set. Then, the depths were binned by elevation to assess similarities in the vertical relief. Next, a relative pixel-by-
pixel difference comparison between the two data sets, calculated by subtracting the $SD_{SfM}$ from $SD_{ASO}$, included the mean,
median, standard deviation, and normalized median absolute deviation (NMAD; Höhle and Höhle, 2009). Finally, the snow
volume was computed for the full watershed and by different surface classifications (snow, rock, vegetation). We note that
the water class from the imaging spectrometer was mostly misclassified shading within vegetation, which we confirmed with





a subset of all water classified pixels. Therefore, we treated both categories (water and vegetated areas) as one category. Additionally, scaling the ASO snow depth map from 3 to 1 m and the fact that the ASO imaging spectrometer does not spectrally unmix pixels to fractional cover resulted in rock classified pixels in the snow-on data set that had measured snow depth. Overall, the focus was to check for similarity in distribution pattern and volumetric agreement for SfM relative to

ASO.

Negative SfM snow depth values, treated as SfM reconstruction and/or co-registration errors, were inspected by terrain characteristics (elevation, slope, and aspect) for the full domain, and by surface classification type from the imaging spectrometer. Aspect and slope were calculated from snow-free lidar acquisition by ASO to create independence from the modeled values by SfM. Median and NMAD for stable terrain differences, using overlapping areas between the SfM snow-

free and snow-on DEM, determined the relative error of the two models.

## 5. Results

### 5.1 Co-registration

Control surfaces, used for co-registration of the SfM snow-free and snow-on scene to the lidar reference point cloud, encompassed 13.9% of the watershed boundaries when gridded at the 1m resolution. The snow-free point cloud was shifted

0.02 m to the North, -0.20 m to the East and -0.41 m in the vertical direction, while the snow-on was 0.01m to the North, -0.02 m to the East and 0.01 m in the vertical. After applying the translation, the differences over the control surfaces in the raster products exported from the respective SfM point clouds had a mean of 0.02 m with a standard deviation of 0.52 m, and median of 0.03 m, with a NMAD of 0.22 m (Figure 2). The remaining difference in the NMAD indicated that there were still some outliers in the control surfaces, despite all the refinements to constrain those. With median and mean close to zero,

however, the co-registration can be considered successful for the two scenes. The NMAD can also be used as a measure for uncertainty in the snow depth values calculated from the two SfM DEMs.

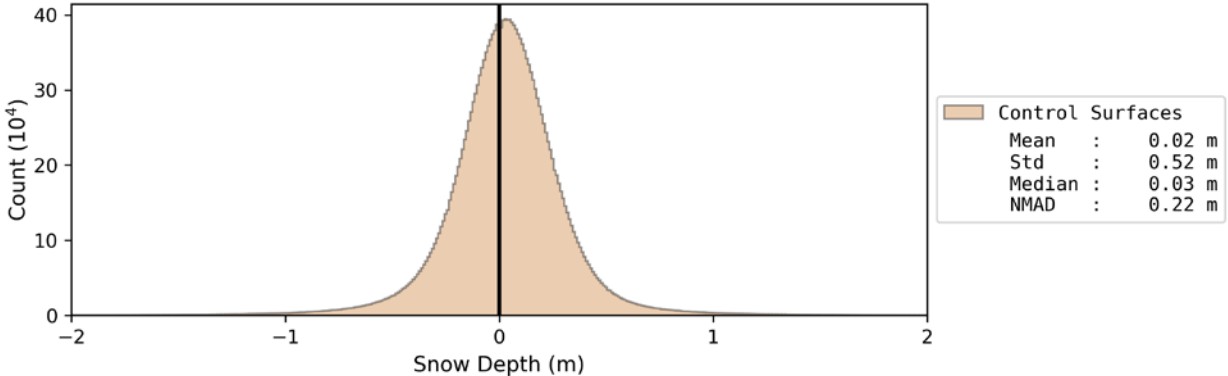

**Figure 2 - Histogram showing the control surface elevation differences subtracting their elevations in the snow-free DEM from the snow-on DEM. The mean difference of 0.02m indicated a successful alignment.**

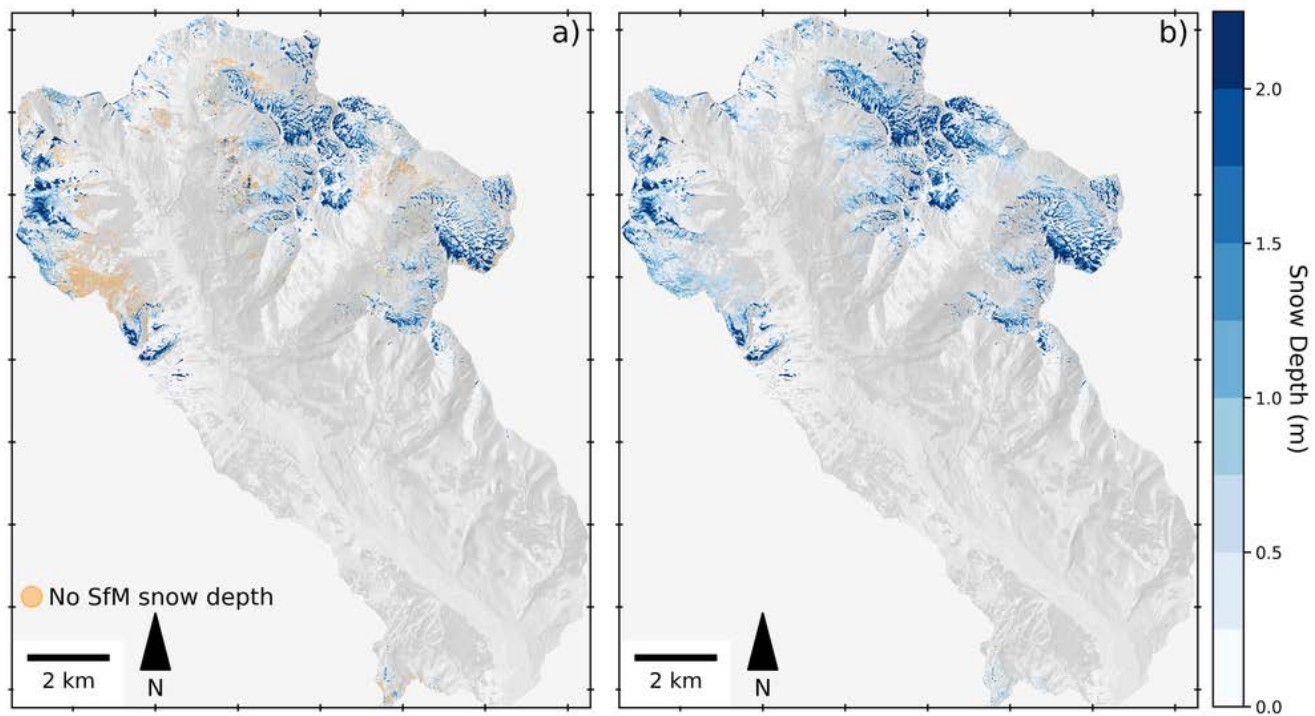

**Figure 3 - Overview of reconstructed snow depth by SfM (a), with ASO snow depth map shown on the right (b). Areas with unsuccessful SfM measurements (orange) coincide with surfaces classified as vegetation or shallow snow depth values (< 1m) by ASO. The snow depth pattern between the two products show good agreement over the overlapping area.**

### 5.2 Structure from Motion snow depth

Overall, where snow depths were measured in both the SfM and ASO snow depth data sets, hereafter referred to as 'SfM' and 'ASO' respectively, there was good agreement in both snow depth and snow volume (Figure 3). Notably, agreement was

best for deeper snow (> 1m) and across higher elevations. There were few gaps, indicating the image sampling configuration was of high enough quality, and with sufficient overlap, to provide a reliable source for reconstruction by SfM. Mean snow depth measured by ASO was 0.89 m, with a median value of 0.64 m, and standard deviation of 0.88 m for the entire domain (Figure 3b). Coinciding SfM snow depths, which covered less snow mapped area by ASO (72%), had a mean of 1.06 m, median of 0.76 m, and standard deviation of 1.11 m (Figure 3a). Most of the spatial coverage difference was caused by non-

positive difference measurements in areas classified as vegetation by the spectrometer or over shallow snow depths (measured from ASO). Snow depth differences in the overlapping area between ASO and SfM, had a mean of 0.01 m, median of -0.03 m and standard deviation of 0.83 m.

Where snow is mapped by both ASO and SfM, there is a very close match in snow volume, with SfM having 1% higher snow volume or a total of $21.10 \times 10^6$ m$^3$. This estimated snow volume was 86% of the ASO measured snow volume for the

entire watershed, which translated to a difference of $3.42 \times 10^6$ m$^3$ of the total $24.52 \times 10^6$ m$^3$. Across land surface



classifications snow depths were mostly mapped by SfM in pixels classified as open snow (81%), followed by rock (12%), and then vegetation (7%). In these land surface categories, SfM underestimated snow volume relative to ASO in the pixels classified as snow capturing 92%, while overestimating snow volume in the rock and vegetation pixels. For ASO, snow volume was distributed differently across land surface types; 69% in open snow, 15% in rock, and 16% in vegetation,
indicating in part that SfM is less likely to successfully map snow in vegetation. An overview of the volume and depth statistics is given in Table 2.

As a whole, SfM showed an underestimation of snow depth compared to ASO (Figure 4a). The depth distribution was mostly in the 0 to 5 m range (Figure 4b), and higher values were more dispersed and highly localized. Other studies that have used ASO snow depth maps also observed extreme outliers and considered these as spurious snow depth values (5 m
McGrath et al., 2019; 6 m Brandt et al., 2020). For this study, we did not remove any high outliers from both data sources and included them in all comparisons.

The distribution of snow depth values across 10 m elevation bands showed higher accumulation in the upper elevations, but not necessarily at the highest elevations (Figure 5) in both data sets. SfM and ASO had increasing depths between ~3200 m and ~ 3800 m, and depths started to decrease again above this range. The values from SfM, however, had a higher spread in
the elevations between 3200 m and 3500 m (Figure 5a), where 74% of the watershed was classified as vegetation. This greater noise pattern was within the expectation, as vegetated areas remain a challenging environment for SfM to measure snow depth (Harder et al., 2020). As a whole, the agreement between the two distributions shows that SfM can be used to map snow depth patterns across a range of elevations in complex terrain.

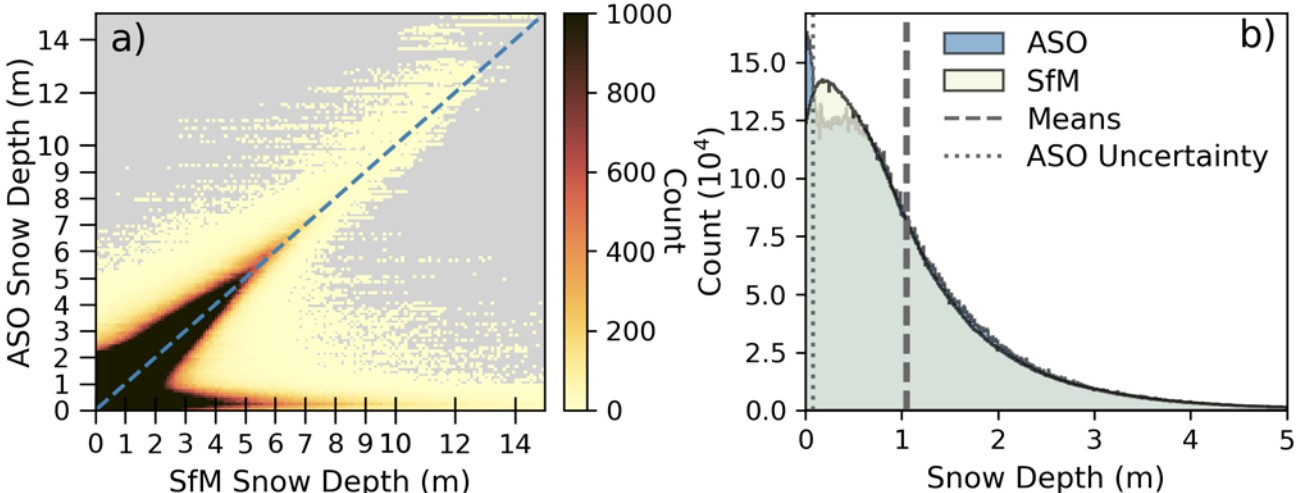

**Figure 4 - SfM snow depth values plotted against ASO snow depth values (a). The dashed line shows a hypothetical one-to-one relationship. SfM tended to underestimate the snow depth compared to ASO. The snow depth histograms showed a strong agreement between the two sources (b).**





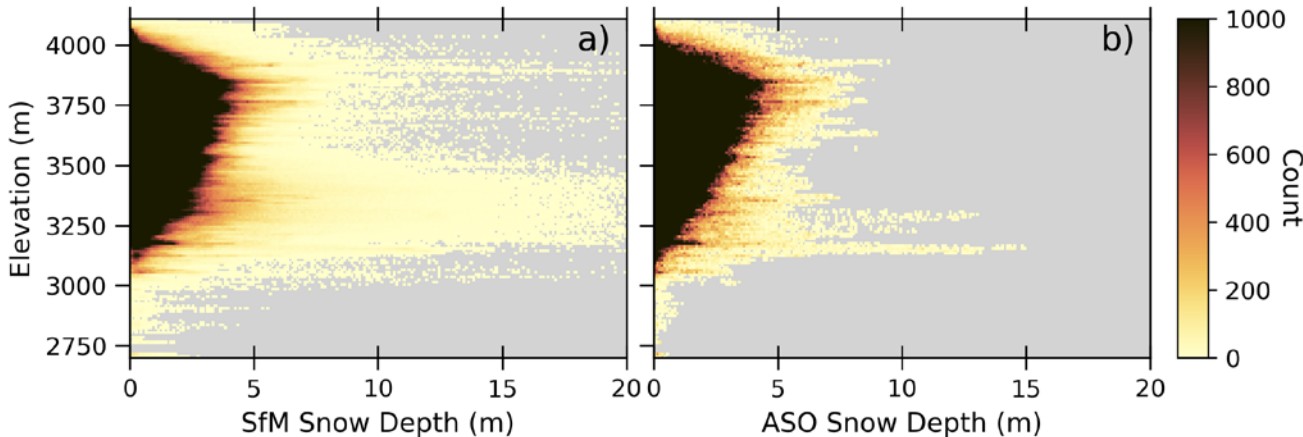

**Figure 5 - Snow depth distribution across elevation bands of 10 m for SfM (a) and ASO (b). Both had similar patterns of higher depth values in the upper elevations and between 3600 m and 4000m. The higher snow depth spread in the lower elevation between 3200 m and 3500 m by SfM is attributed to more areas with vegetation.**

**Table 2 - Overview of snow volume and depth statistics for SfM and ASO.**

|  | **SfM** | **ASO** | *Difference* |
|---|---|---|---|
| Total Snow Volume (m$^3$) | 21.10 x 10$^6$ | 24.52 x 10$^6$ | -3.42 x 10$^6$ |
| In pixels with SfM and ASO depth | | 20.84 x 10$^6$ | 0.26 x 10$^6$ |
| Snow Covered Area (SCA) | 72% | 100% | 28% |
| Mean Depth (m) | 1.06 | 1.05 | 0.01 |
| Median Depth (m) | 0.76 | 0.79 | -0.03 |
| Standard Deviation (m) | 1.11 | 0.96 | 0.83 |

*Note: Mean, Median, and Standard Deviation are for overlapping area by SfM.*

**5.3 Structure from Motion measurement errors**

The area mapped as snow by ASO but not by SfM spanned 28% of the snow-on area present in the ASO depth map, of which 0.5% was a data gap in SfM. The snow depth that was 'missed' by SfM had a mean of 0.48 m, median of 0.39 m, and a standard deviation of 0.42 m. As a simple exercise to estimate the missed amount of snow water equivalent (SWE), we applied a constant snow density of 350 kg/m$^3$ to the mapped area from ASO and SfM. Using this estimate, the result for

ASO matched within 4% of the official SWE reported by ASO for this flight, which is based on a pixel-wise modelled density. The total water amounts were 8.5 x 10$^6$ m$^3$ (ASO) and 7.3 x 10$^6$ m$^3$ (SfM), resulting in a difference of 1.1 x 10$^6$ m$^3$.

To further investigate the pixels with no measured depth from SfM, we compared them to the corresponding snow depth values from ASO (Figure 6). This showed that SfM failed to map snow where ASO mapped shallower snow (<1 m; Figure 6a) in open areas, and the largest negative SfM values (-5 m and -28 m) were primarily found in areas classified as



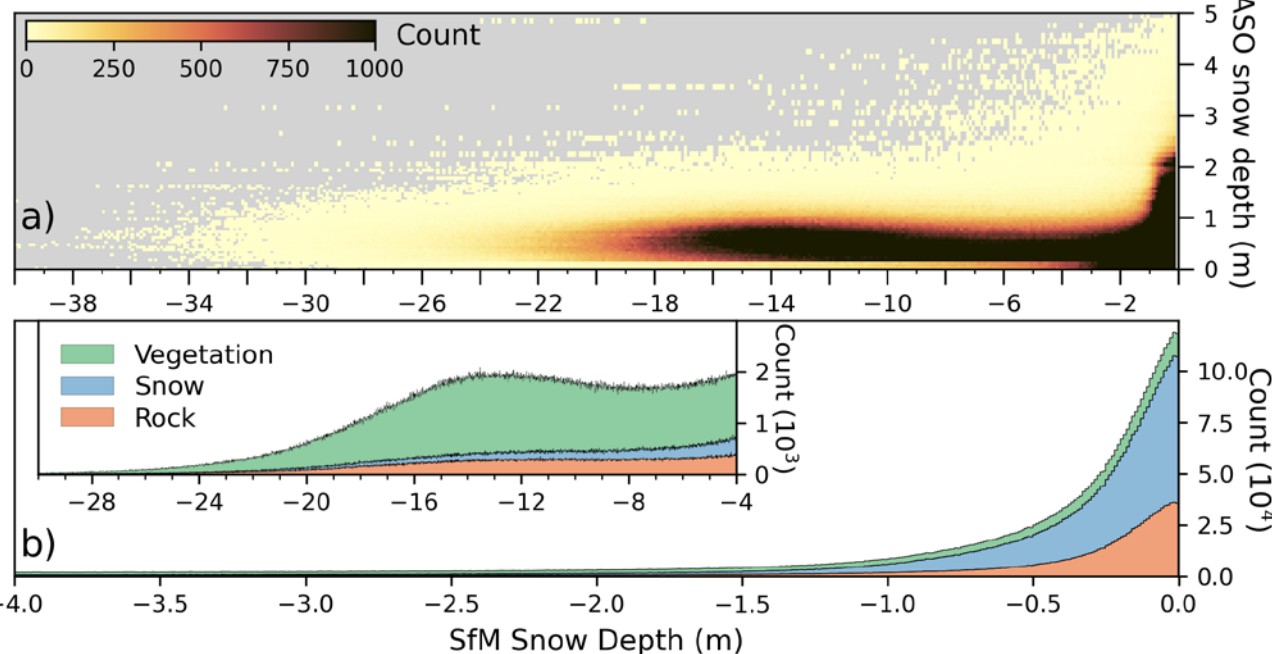

**Figure 6 - Snow depth with negative values by SfM plotted against values by ASO. Extreme outliers are dominantly found in areas with snow depth of less than 1 m (a) or vegetated areas (b).**

vegetation (Figure 6b). As previously mentioned, vegetation is challenging for SfM and these results are in line with previous SfM work, where shallow depth or forested areas were shown to impair the ability of SfM to measure accurate surface elevations (Avanzi et al., 2018; Fernandes et al., 2018).

An analysis to correlate areas of snow missed by SfM with terrain characteristics showed no strong relationships for the area as a whole, or across land surface classification types. Out of the investigated influence factors of aspect, elevation, and
slope, the most visible trend was detected when values filtered to only open areas (no vegetation) were binned by slope angle and the median depth calculated. The median did not exceed -1 m and showed a linear trend up until around 55 degrees, then

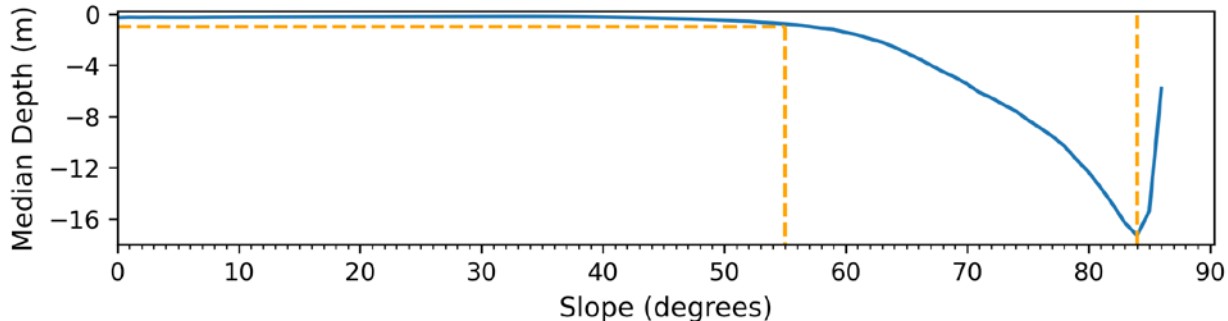

**Figure 7 - Median snow depth binned by slope angles showed a linear trend until 55 degrees and stayed below -1m (orange dotted lines), before increasing sharply.**




started to decrease sharply (Figure 7). This observation is similar to other studies, where slopes above 50 degrees show a decline in accuracy for photogrammetric reconstructions (Shean et al., 2016; Shaw et al., 2020).

## 6 Discussion

### 215   6.1 Structure from Motion with airplane imagery

The primary focus of ASO is the delivery of lidar-based snow depth and snow water equivalent maps, and the camera images are not currently used as a resource in data product processing (Painter et al., 2016). With the lidar and imaging spectrometer as the primary data streams, there is little consideration given to the image overlap, illumination conditions for the camera sensor, or minimum GSD for further use with photogrammetric reconstruction. Given this image acquisition setup and the

presented results, we believe that although the results here are promising, there is room for improvement if flight campaigns were planned to produce snow depth maps with SfM. For instance, consistent image overlap can improve the quality of SfM output products (Bühler et al., 2016; Harder et al., 2016; Meyer and Skiles, 2019). The potential for snow-depth mapping by SfM has been demonstrated on a smaller scale by Nolan, et al. (2015) with an accuracy of +/- 0.3 m. Our NMAD of 0.22 m over a larger target area denotes the scalability of this technical setup and is in line with Eberhard, et al. (2020), where the

NMAD was 0.17 m. Another indicator for the capability of SfM is shown by the point density of the two SfM point clouds. Here, we had an average of 23.2 points/m$^2$ for the snow-on acquisition and 31.5 points/m$^2$, which signifies well reconstructed surfaces by SfM. With the high point density, higher resolutions for the gridded output products are possible, as well.

We acknowledge that the combination of reference and comparison data with a single acquisition is unique to ASO's

recording setup. The data set provided an opportunity to perform a comparison of SfM to an established snow depth mapping technology, but we note that it is not needed to perform this methodology outside of the ASO operation domains. For classification, the snow-on and snow-free point clouds can be directly classified using the SfM point clouds and the image RGB information (Shaw et al., 2020) or near-infrared spectrum (Deschamps-Berger et al., 2020), where available. Producing the classification with this approach was beyond the scope of this work and warrants an accuracy assessment by itself before

continuing to use in downstream products. Using the existing classification, we reduced a potential source of error for the depth and volume assessment. Once classified, ensuring proper geo-location of the models can be completed by co-registration against suitable control surfaces from any externally sourced point or gridded based referenced data set and solutions to complete this already exist (Shean et al., 2016). For areas with little change to control surfaces (exposed rock surfaces or roadways), the reference DEM can further be from different recording years and does not have to be acquired

within the same year of the images (Midgley & Tonkin, 2017). In the end, the presented processing steps can be applied to any airborne collected and geo-referenced image data set. A lidar-based reference or image spectrometer classification is not required.



## 6.2 Absence of Ground Control Points

Ground control points (GCP) are commonly used for RPAS based studies to geo-reference their results, which strongly
influences accurate geo-location (James et al., 2017). Our process explicitly precluded the use of GCPs to reduce manual
processing intervention and increase automation potential. We believe that image geo-location and perspective information
combined with co-registration is a reliable substitute to GCP's, while not compromising on output quality. Co-registration is
a common practice for photogrammetric snow depth products from satellite image (Shean et al., 2016; McGrath et al., 2019;
Shaw et al., 2020; Deschamps-Berger et al., 2020) and equally applicable for areal imagery across larger alpine areas. The
SfM software performed very well with the image metadata and the snow-on model was very close to the lidar, while the
snow-free model had a higher shift, predominantly in Y (-0.20 m) and Z (0.41 m) direction. We hypothesize that the snow-
free scene, with more exposed vegetation and ground cover, degraded the accuracy for SfM. With both alignment
adjustments very low in magnitude, it is further feasible to align the two models to each other and compute snow depth and
volume in relative geo-location space. Alpine areas benefit from having exposed control surfaces for multi-view image
processing and co-registration, having identifiable features in both scenes. For different environmental conditions, such as
ice-sheets that have little to no overlapping stable terrain, alternative approaches have been developed to align corresponding
surfaces (Howat et al., 2019; Shean et al., 2019).

## 6.3 Comparison to other platforms

The SfM NMAD from airplane imagery in this study shows a higher accuracy compared to satellite-based stereo
photogrammetric studies, where the NMAD ranges from 0.36 m (Shaw et al. 2020) over 0.45 m (Marti et al. 2016) and up to
0.69 m (Deschamps-Berger et al., 2020). Reasons for the higher accuracy can be topographical, as satellite stereo pairs have
a larger area with more varying terrain in a single scene, which makes it more difficult to capture high enough detail of
information for reconstruction. Additionally, DEM generation from satellite images has a different technical setup, where
stereo photogrammetry uses up to three images (tri-stereo) (Shaw et al, 2020, Deschamps-Berger et al., 2020, Bhushan et al.,
2021). This varies for SfM, which can use any number of images, driven by the amount of overlap in an area. Weather
conditions are an additional high impact factor when using satellite imagery. An unobstructed view from an instrument to the
entire study area at the time of overpass cannot be guaranteed, and atmospheric features like clouds can cause additional
occlusions. On the smaller scale using RPAS platforms, the accuracy is higher (cm scale) compared to what we have
achieved here (Avanzi et al., 2018, Harder et al., 2016, Bühler et al., 2016). This can be attributed to the lower flight altitude
and resulting higher degree of image overlap and GSD. One of the remaining challenges for RPASs, though, is the ability to
cover larger areas with limited battery life, higher sensitivity to weather conditions in alpine areas, and access challenges to
operate safely (Bühler et al, 2016).

Given these limitations, we see piloted aircraft SfM filling an important gap between RPAS-SfM and satellite stereo
photogrammetry. The reported accuracy for ASO snow depth maps at the 3 m resolution is 0.05 m (ASO, personal



communication). Given the higher resolution of the SfM outputs and the accomplished NMAD here, we believe that SfM compares well against an active measurement instrument like lidar on larger scales. This assessment holds for open spaces with a snowpack of more than 1 m. As with other studies, we see vegetated areas and shallower depth a remaining challenge for SfM, and the technology still needs to be improved in order to match results that are possible with lidar (Harder et al., 2020). Steep terrain is another aspect where accuracy for the results degrades, particularly on angles above 50 degrees. Here,

we argue that the accumulation in those areas is low and also see a need in the literature to assess the quality of remote sensing measurements.

## 6.4 Expanding snow science

The ability to fill missing information between point-based snow depth measurement locations has improved our understanding of large-scale snow processes. Data sets from airborne campaigns have been used to improve model

capabilities to predict snow precipitation (Behrangi et al., 2018), observe snowfall distributions (Brandt et al., 2020), or improve snow energy balance models (Hedrick et al., 2020). Expanding the number of observed regions with spatially and temporally extensive records can further accelerate our ability to understand snow processes at scale. Although SfM is not yet able to deliver similar accuracies to lidar for all terrain characteristics and land cover classes, it can be used to supplement or build upon lidar data sets. For instance, a first survey could be conducted using the more accurate lidar, and

successive observations use the more cost-efficient SfM for open areas with little forested areas (Pflug and Lundquist, 2020). With the results of this work, we demonstrate that SfM can be an option for operations like ASO for repeated observations after the initial lidar flight. From a technical setup perspective, it is further feasible to source the images from space-borne platforms, adding the option of temporally consistent broad-scale coverage and reduce operational requirements.

## 7. Conclusions

This study's motivation was to investigate whether Structure from Motion should be considered an additional remote sensing data source for snow depth monitoring on a large watershed scale. It also emphasized to keep the manual intervention for data processing to a minimum to be scalable with area size. The results for depth and volume compared to a co-incidental ASO lidar-based measurements at a 1m resolution showed almost identical statistics for mean, median, and standard deviation for depth, and a slight overestimation in volume. The co-incidental surfaces in open areas matched 101.26% in

volume and a slight difference of 0.01 m in the mean snow depth of 1.06 m. As with previous studies, vegetated, steep, or shallow snowpack areas had high reconstruction errors, with no measured snow depth. These terrain and snow depth characteristics accounted most for the missed volume by SfM compared to the ASO snow depth product.

We would like to see Structure from Motion applied to larger areas and more frequent image acquisition to improve our understanding of this technology at scale. As with lidar, it can provide high resolution spatially complete data sets with sub-



meter accuracy. This capability can further improve our ability to model and understand snow-driven hydrological processes and contribute to explaining the consequences of our changing environment.

**Author contributions**

JM and MS conceptualized the overall study, with helpful contributions from JD and DS on parts. JM performed the data processing and analysis. KB provided ASO data and support. MS provided financial support for the study. JM wrote the first
draft of the manuscript, which was then contributed to by all authors.

**Code availability**

The software components used to process data are publicly available on https://github.com/UofU-Cryosphere/snow-aso and analytical code for the output data is available on: https://github.com/jomey/raster_compare

**Data availability**

Camera images, lidar point cloud files, and imaging spectrometer ground classification were acquired through personal communication with the ASO team. The snow depth map is publicly available through the National Snow and Ice Data Center Distributed Active Archive Center download portal (https://nsidc.org/data/ASO_3M_SD/versions/1).
Additionally, the processing reports from Agisoft Metashape have been provided in the supplemental material.

**Competing interests**

Co-authors Jeffrey Deems and Kat Bormann were members of the NASA ASO team (which produced the data used in this study). Jeffrey Deems is a co-founder of Airborne Snow Observatories, Inc. and Kat Bormann is currently employed by ASO, Inc., formed as a result of the ASO NASA technology transition effort.

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
