# Peer review of "Mapping snow depth and volume at the alpine watershed scale from aerial imagery using Structure from Motion"

_The Cryosphere, 2021_

## Referee Comment (RC1)

Review of *Mapping snow depth and volume at the alpine watershed scale from aerial imagery using Structure from Motion* by Joachim Meyer et al.

General Comments

The manuscript by Meyer et al. explores the applicability of airborne structure-from-motion (SfM) to derive high spatial resolution DEMs and snow depth through DEM differencing in an alpine watershed. The authors use the coincident in time LiDAR information from the well-established Airborne Snow Observatory (ASO) to evaluate the SfM approach and explore the errors for different topographic and surface characteristics. At a 1 m posting, the authors report small, decimetre errors relative to ASO LiDAR and support existing findings that SfM fails to capture snow depth variations for shallow snowpacks (<1m), in vegetated areas or on steep slopes. The paper is well written, to the point and produces some nice additional evidence of SfM as a tool for deriving snow depth in complex terrain when compared to highly detailed reference dataset. While I do not have any major concerns with the science presented in the current manuscript, I also feel that the authors could do a little more to advance the current state-of-science through some small additional analyses, figures and discussions.

My main point is regarding the future potential of large airborne SfM campaigns that could be explored and discussed a bit further in the manuscript. As the authors acknowledge, a coincident LiDAR and SfM dataset is extremely unique and is likely to be beyond the capabilities of many studies who might wish to study catchment-wide snow depth/SWE variations. Accordingly, I think the manuscript would benefit nicely from two things. Firstly, I would like to see some further discussion on likely future pathways for mapping snow depth where ASO or detailed airborne flight lines are not available. The authors suggest that a LiDAR survey can be established on one occasion and used to aid successive SfM surveys (L296-297) but in reality, the cost and wide availability of flyovers still restrict its application in many domains. Second, with such a dataset as this, the authors should attempt to extend the analysis of the residuals against a more detailed classification of vegetation types (if grassland and coniferous forests are all classed as one 'vegetation' class but likely give different results for different reasons) and other topographic characteristics (terrain roughness, curvature or exposure to wind for example). As this SfM-LiDAR dataset is so unique, I think it should be leveraged to really test the limits of SfM. I added other suggestions for the discussion (specific comments below) that I believe the authors could expand upon and even provide small tests to support.

Finally, I think the authors could improve on a few figures in order to aid the reader and maximise the assessment of the SfM-LiDAR in this paper. For example, providing more detail on the map(s) to show where the ground control/vegetation areas are.

Specific Comments

Abstract:

L15: It's well known by the cryospheric community at this point, but I would still add the definition of 'ASO' and 'SfM' for the reader in the abstract.

L16: Remove 'though' after ASO.

L16: It is not clear from the abstract how SfM maps less snow coverage. This is due to negative DEM differencing for thin snowpacks and thus classed as zero snow? Perhaps this sentence could be reformulated to clarify somehow.

L24 (plain language summary): The authors themselves do not explore continental scales, so this is misleading as a background summary.

Introduction:

L29: Reword "observables"

L31: It is not clear to me what "mid-elevations index sites" are. Could the authors clarify this or add a reference?

L37: "resolutions"

L38: These platforms don't have a defined accuracy so this a little misleading to read. Perhaps "….and resulting in varying degrees of accuracy for monitoring snow depth." Or something similar.

L41: "photogrammetry-derived"

L42-44: Restructure sentence and improve syntax.

L44: This reference is perhaps a bit misleading with the sentence. Shaw et al. benefitted from using Pléiades in central Chile, where there are few clouds in spring-summer. Please reformulate (as above comment).

L46: A few references here would be reasonable. Not just for ASO, but other airborne campaigns for looking at snow.

L47: LiDAR mentioned for the first time. Again, this is well established by now, but should be defined: Light Detection and Ranging (LiDAR / lidar).

L51: "up to alpine catchments size" is very vague and could encompass a huge range of area. Please specify.

L54: I would also add some considerations to pilot line of sight and controller signal etc.

L63: I would add Bühler et al., 2015 as a reference here too.

L65: hyphenate "SfM-based".

Study Area:

L75: Write 'Colorado' in full for readers less accustomed to the state abbreviations.

L75: Add catchment centre coordinates to the text line (and coordinates for the map in Figure 1).

L83: It would be useful, especially given the assessment of vegetation in the manuscript, to highlight where the vegetation is on the map in Figure 1.

Data:

L93: provide as ordinal numbers (e.g. 24$^{th}$ and 12$^{th}$)

L99: Do the authors find anything relevant to report with respect to the impact of image saturation, lighting etc and how it might affect the image correlation for SfM?

Methods:

L116: Remove "interested". Everyone is interested 😊

L125: It would again be useful to the reader to be able to interpret where these control surfaces are, along with their spatial distribution. The reader has no way to be sure that the spatial distribution of control points is well distributed in space.

L126-128: Again, the reader cannot gain an appreciation for how much extension of the boundary there is and how this relates to additional control ground.

L130: I think the authors need to justify their decision to explore a 1m posting. Were other resolutions tested. It has been established that errors are reduced when spatially averaging to coarser resolutions (e.g. Deschamps-Berger et al., 2020) and that practically, any application to modelling water resources with this data would likely not use a 1 m resolution. For example, most studies like this have applied models at ~25-60 m resolutions (Margulis et al., 2019; Pflug and Lundquist, 2020; Shaw et a., 2020(b)). I think there should be some consideration of this and, better still, some presentation of results for different resolutions.

L133: Please provide a mean point density.

L139: Please add a reference(s) here. Painter et al. etc

L139: Reword "distributes". Perhaps "outputs".

L139-141: See above point regarding posting resolution and its justification.

L146: ".. and grouped by the different underlying surface classifications (snow, rock, vegetation)."

L148: I'm not convinced that this is an appropriate grouping of water bodies and vegetated areas. It has for sure been demonstrated that SfM techniques struggle with mapping snow beneath trees. For water bodies though, snow on top of frozen ponds or lakes in the winter scene would provide a reasonable DEM result, but the 'snow depth' will be affected more by the noise generated from the snow-free imagery where there is movement of the water, for example. Especially as the authors give some attention to the SfM approach for vegetated areas, it seems inappropriate to group them together as they will likely give poor results but for different reasons. Moreover, the classification of vegetated areas supposedly groups together small bushes, grassland and larger conifer trees (as suggested in section 2)? I think that the results of SfM vs ASO should be explored separately for those classes to strengthen the assessment of SfM's merits. I would also like to see the water bodies added as a classification on the map as the reader has no way of understanding their importance for this work.

L153-157: It was not made clear to me whether the authors removed the vertical biases of the snow-free and snow-covered SfM DEMs after co-registration with ASO DEMs. To appropriately derive the snow depth from DEM differencing, this vertical bias should be removed using the aforementioned un-vegetated stable ground of <50° (assuming it is trustworthy and well distributed). If this bias has been removed, then the median should be zero anyway and doesn't need reporting.

Results:

L165: See above point regarding median differences.

L176-178: Can the authors clarify and reword this sentence with respect to "non-positive difference measurements", as this is not clear. You mean to say that all differences over vegetation areas are negative because SfM adds the noise from trees?

L178: This is not clear. What overlapping area? All the area covered by snow in both ASO and SfM maps?

L180-188: Please restructure this paragraph as the percentage differences (e.g. L181, L185) and area percentages affect the flow of the text and could be made clearer.

L190: How are the higher snow depth values highly localised? Can the authors clarify in the text?

L191-192: "5 m in McGrath et al., 2019 and 6 m in Brandt et al., 2020)."

L197: ".. with 74% of that elevation band classified as vegetation."

L198: "..was expected.."

L204: Can the authors elaborate here (or earlier) why there were data gaps for the SfM when it uses the same path of the LiDAR? Was this a saturation/image correlation issue/impact of vegetation?

L206: Is 350kg m3 and appropriate density for a mid-May snowpack in Colorado? It appears a little low to me. Admittedly, the authors suggest that this is a simplification, but I'm wondering whether a more appropriate value could be derived/taken from the literature. Or was this taken from the mean of modelled density?

L207: remove "official".

L209: Again, not clear. How was no snow 'measured' from SfM here? Because it was a negative DEM difference? Please state this in the text if so.

Discussion:

L225: I think that the image quality and illumination is an area that could be tested for the SfM data here if it has any relevance. If it doesn't, perhaps it could be stated?

L233: Pease add point density earlier in the text.

L236: Not clear. You mean a space-time coincident LiDAR and SfM product? Please re-write.

L250-264: This section is generally fine, but it made me think about doing some further testing with regards to the domain and the available stable terrain. When domains are small for RPAS measurements, the distribution of stable points is even more limited and reliance on GCPs to align and evaluate the DEMs and avoid 'bowl effects' are even more crucial. What if the authors used smaller subdomains with more limited bareground for co-registration? I think some additional discussions related to this could be useful for the community, especially given the unique and special dataset the authors have that is not normally available (see major comment).

L261: Why was the snow-free SfM scene hypothesised to result in a degraded accuracy? Due to more irregularities over the surface that are smoothed by snow coverage? Does this outweigh the potential loss of correlation over smooth surfaces with little contrast between pixels? A bit more discussion on this could be good.

L261-262: Perhaps, but many of these surfaces are often of steep slope and therefore not appropriate for co-registration or bias removal. They are also not often well distributed in space (again a map showing this would be nice) and are likely to be clustered at lower elevations which are warmer (patchy snow) and shallower in slope.

L276: Adding Goetz and Brenning (2019) would also be appropriate here.

L284: I agree that the authors see a rise in the residuals below 1 m (or even 0.6-0.7 m), but I'm not certain that the approach is bad for capturing snow < 1m. Again, the error also drops as one coarsens the resolution of the DEM/snow depth map (Deschamps-Berger et al., 2020).

L286-288: Another interesting discussion point is the comparison of the airborne (ASO) SfM to the ASO LiDAR and the Pléiades imagery in Tuolumne where this work has already been started (Deschamps-Berger et al., 2020). I'm surprised this manuscript did not also explore the Tuolumne for that reason. Could this be tested by the authors to show the differences between ASO-based SfM and Pléiades? Or at least hinted at in this discussion maybe?

L293: Also estimate SWE (Margulis et al., 2019) and streamflow (Shaw et al., 2020(b))

Conclusions:

L306: This is the first time I see this number reported and it's not clear what a match of 101.26% in volume means.

L308: I'm not sure that <1m is a "shallow snowpack", especially not considering the total range of reported snow depths (i.e. Figure 3).

Figures

Figure 1: Please add a north arrow to the map as well as providing some coordinates. I would consider adding contours or some elevation-related information to the map. Please consider adding land cover classifications and stable ground areas (even as a subplot) as well as they are emphasised quite a lot in the text.

Figure 3: I think that a residual map would be particularly informative here.

Figure 4: Can the authors zoom on the lower snow depths on panel b) somehow. The information there is particularly interesting but not so clear. Maybe ASO could also be plotted on top?

Figure 5: The authors give mention to the distribution of vegetation with reference to this figure (L197). The authors should add a simple vegetation distribution plot against elevation using a percentage of the total in each elevation band used for the snow depth stats.

Figure 6: This is a nice figure. I'm curious why only negative differences are shown here. I think the authors could nicely leverage a residuals map plotting against all surface topographies (slope, elevation, aspect, roughness) and classifications (trees, grassland, water?, rock, bare ground).

Figure 7: Is this actually a median snow depth difference? I think that any 'snow depths' supported on slopes above 70° could not be detected from SfM anyhow. I would cut to 0-70° and show "DEM difference" for both snow-covered pixels and bare ground pixels. The authors

could expand this and show the different surface classifications against slope here too. What about the ASO 'snow depths' Until what slope angle can they be considered trustworthy?

Tables
Table 1: I think that this table is unnecessary given the full detail provided in the text (cf. L100-106).

Cited Literature

Bühler, Y., Marty, M., Egli, L., Veitinger, J., Jonas, T., Thee, P., and Ginzler, C.: Snow depth mapping in high-alpine catchments using digital photogrammetry, The Cryosphere, 9, 229–243, https://doi.org/10.5194/tc-9-229-2015, 2015.

Deschamps-Berger, C., Gascoin, S., Berthier, E., Deems, J., Gutmann, E., Dehecq, A., Shean, D., & Dumont, M. (2020). Snow depth mapping from stereo satellite imagery in mountainous terrain: Evaluation using airborne laser-scanning data. *Cryosphere*, *14*(9), 2925–2940. https://doi.org/10.5194/tc-14-2925-2020

Goetz, J., & Brenning, A. (2019). Quantifying uncertainties in snow depth mapping from structure from motion photogrammetry in an alpine area. *Water Resources Research*, 55, 7772–7783. https://doi.org/10.1029/2019WR025251

Pflug, J. M., & Lundquist, J. D. (2020). Inferring Distributed Snow Depth by Leveraging Snow Pattern Repeatability: Investigation Using 47 Lidar Observations in the Tuolumne Watershed, Sierra Nevada, California. *Water Resources Research*, *56*(9), 1–17. https://doi.org/10.1029/2020WR027243

Margulis, S. A., & Fang, Y. (2019). The Utility of Infrequent Snow Depth Images for Deriving Continuous Space - Time Estimates of Seasonal Snow Water Equivalent Geophysical Research Letters. *Geophysical Research Letters*, *46*, 5331–5340. https://doi.org/10.1029/2019GL082507

Shaw, T. E., Caro, A., Mendoza, P., Ayala, Á., Pellicciotti, F., Gascoin, S., & McPhee, J. (2020(b)). The Utility of Optical Satellite Winter Snow Depths for Initializing a Glacio-Hydrological Model of a High-Elevation, Andean Catchment. Water Resources Research, 56(8), 1–19. https://doi.org/10.1029/2020WR027188